# Short-chain ketone production by engineered polyketide synthases in *Streptomyces albus*

Satoshi Yuzawa [1,2,11], Mona Mirsiaghi[1,3], Renee Jocic[1], Tatsuya Fujii[2,4], Fabrice Masson[1,3], Veronica T. Benites[1,2], Edward E.K. Baidoo[1,2], Eric Sundstrom[1,3], Deepti Tanjore[1,3], Todd R. Pray[1,3], Anthe George[2,5], Ryan W. Davis[5], John M. Gladden[2,5], Blake A. Simmons [1,2], Leonard Katz[2,6] & Jay D. Keasling [1,2,6,7,8,9,10]

Microbial production of fuels and commodity chemicals has been performed primarily using natural or slightly modified enzymes, which inherently limits the types of molecules that can be produced. Type I modular polyketide synthases (PKSs) are multi-domain enzymes that can produce unique and diverse molecular structures by combining particular types of catalytic domains in a specific order. This catalytic mechanism offers a wealth of engineering opportunities. Here we report engineered microbes that produce various short-chain (C5–C7) ketones using hybrid PKSs. Introduction of the genes into the chromosome of *Streptomyces albus* enables it to produce $>1\,g\cdot l^{-1}$ of C6 and C7 ethyl ketones and several hundred $mg\cdot l^{-1}$ of C5 and C6 methyl ketones from plant biomass hydrolysates. Engine tests indicate these short-chain ketones can be added to gasoline as oxygenates to increase the octane of gasoline. Together, it demonstrates the efficient and renewable microbial production of biogasolines by hybrid enzymes.

[1] Biogical Systems and Engineering Division, Lawrence Berkeley National Laboratory, Berkeley, California 94720, United States. [2] Joint BioEnegy Institute, Emeryville, California 94608, United States. [3] Advanced Biofuels & Bioproducts Process Development Unit, Lawrence Berkeley National Laboratory, Berkeley, California 94720, United States. [4] Research Institute for Sustainable Chemistry, Institute for Synthetic Biology, National Institute of Advanced Industrial Science and Technology, Higashi-hiroshima, Hiroshima 739-0046, Japan. [5] Department of Biomass Science and Conversion Technologies, Sandia National Laboratory, Livermore, California 94551, United States. [6] QB3 Institute, University of California, Berkeley, California 94720, United States. [7] Department of Bioengineering, University of California, Berkeley, California 94720, United States. [8] Department of Chemical and Biomolecular Engineering, University of California, Berkeley, California 94720, United States. [9] Novo Nordisk Foundation Center for Biosustainability, Technical University of Denmark, Building 220, Kemitorvet, DK-2800 Kgs, Lyngby, Denmark. [10] Center for Synthetic Biochemistry, Shenzhen Institutes for Advanced Technologies, Shenzhen, Guangdong 518055, China. [11]Present address: Biotechnology Research Center, The University of Tokyo, Tokyo 113-8657, Japan. These authors contributed equally: Satoshi Yuzawa, Mona Mirsiaghi. Correspondence and requests for materials should be addressed to S.Y. (email: usyuzawa@g.ecc.u-tokyo.ac.jp) or to J.D.K. (email: jdkeasling@lbl.gov)

According to the United States (US) Environmental Protection Agency (EPA), global carbon emissions from fossil fuels have increased by >500% since 1950 and reached approximately 10 billion tonnes in 2014 (https://www.epa.gov/ghgemissions/global-greenhouse-gas-emissions-data). The transportation sector contributed about 15% of greenhouse gas (GHG) emissions, with 95% of that coming from petroleum-based fuels. The US industrial sector also contributed approximately 20% of GHG emission from the burning of fossil fuels for energy and the production of commodity chemicals from fossil resources. Fuels and commodity chemicals produced from renewable sources, such as biomass, would significantly reduce GHG emissions.

A US National Bioeconomy Blueprint released by the White House proposed to reduce our dependence on petroleum through biological generation of fuels and chemicals[1]. Microbial fermentation is an attractive approach to produce these compounds renewably. For example, several genetically engineered microbes have been developed to produce fatty acid esters, fatty alcohols, and fatty ketones that could be used as drop-in replacements for diesel fuels or as flavors and fragrances[2–6]. Compared to the traditional biofuel ethanol, higher alcohols offer advantages as gasoline alternatives due to their higher energy density and lower hygroscopicity. Microbial production of C3–C10 alcohols has also been reported in several genetically engineered microbes[7–10]. Hydrocarbons, such as alkanes or alkenes, are of particular interest because they are similar to the petroleum-based fuels currently in use (a mixture of C4–C12 hydrocarbons) and have higher energy content than ethanol. Bio-based production of hydrocarbons has also been reported[11–13].

Short-chain ketones could potentially be used as gasoline replacements or oxygenates in gasoline because of their high octane numbers[14]. However, although some microbes (mainly Clostridium strains) naturally produce acetone, no natural metabolic pathway is known to produce detectable levels of the other short-chain ketones. Recently, E. coli was engineered with a promiscuous β-keto-thiolase from R. eutropha, a coenzyme A (CoA) transferase from Clostridium acetobutylicum, and an acetoacetate decarboxylase from C. acetobutylicum, and produced $1.3 \text{ g} \cdot \text{l}^{-1}$ of butanone and $240 \text{ mg} \cdot \text{l}^{-1}$ of 2-pentanone from the corresponding acyl-CoAs[15,16]. Although this CoA dependent chain elongation pathway, which resembles the reversed β-oxidation pathway, is one of the promising routes to produce methyl ketones, it is not known if the β-keto-thiolase can utilize branched-chain acyl-CoAs to produce other types of methyl ketones or produce ethyl ketones using propionyl-CoA in chain elongation.

Previously, we engineered E. coli to produce low titers ($<5 \text{ mg} \cdot \text{l}^{-1}$) of an ethyl ketone (3-pentanone) or a methyl ketone (butanone) by introducing a hybrid polyketide synthase (PKS)[17–19]. The ethyl ketone-producing PKS was constructed by fusing LipPks1, a PKS subunit of lipomycin synthase from Streptomyces aureofaciens Tü117 (hereafter referred to as S. aureofaciens) and a thioesterase (TE) domain of the erythromycin PKS from Saccharopolyspora erythraea, and by inactivating the ketoreductase (KR) domain of LipPks1. The methyl ketone PKS was created by further engineering the ethyl ketone PKS where the second acyltransferase (AT) domain of LipPks1 that incorporates methylmalonyl-CoA was replaced with an AT homologue that incorporates malonyl-CoA (Fig. 1). These engineered enzymes were capable of producing small amounts of five ethyl ketones and five methyl ketones by condensing different acyl-CoAs and carboxyacyl-CoAs in vitro[17]. To construct a more efficient PKS, we further engineered the ketone PKSs and introduced the genes into several Streptomyces species. Together, these developments enabled production of industrially relevant ketones at greater than $1 \text{ g} \cdot \text{l}^{-1}$ in shake flasks. These results suggest

potential industrial applicability of these microbial platforms to produce short-chain ketones that could be used as a gasoline replacement and/or a gasoline additive.

## Results

**Rational PKS engineering.** Recently, we found that the employment of an alternative translational start site (TSS) in LipPks1 dramatically increased the protein level of LipPks1 + TE and the corresponding product levels, 3-hydroxy acids, in a heterologous Streptomyces host, Streptomyces venezuelae ATCC 10712 (hereafter referred to as S. venezuelae), compared to the originally annotated start codon[20]. Using the newly identified TSS, which truncates a part of the N-terminal tail of LipPks1, we redesigned the ketone PKS genes (Fig. 1). In addition, we sought to change the inactivating mutation of the KR domain from the serine to the catalytic tyrosine since the role of the serine could be partially replaced by the other amino acids based on the mechanisms shown in Supplementary Fig. 1a. We therefore mutated the codon that encodes the tyrosine to that of phenylalanine (SSIAGVWGSGDHGA<u>Y</u>A to SSIAGVWGSGDHGA<u>F</u>A). The resulting ethyl and methyl ketone PKS genes (Fig. 1 and Supplementary Data 1) were introduced into the chromosome of four Streptomycetes, and short-chain ketone-production levels were determined as discussed below.

**Streptomyces host selection.** Natural products from the genus Streptomyces have been the source for many clinically useful pharmaceuticals. In fact, over one-third of known polyketides and nonribosomal peptides originate from this genus[21]. Successful heterologous production of natural products of Streptomyces origin has mostly been achieved in well-characterized model Streptomyces hosts such as S. coelicolor, S. lividans, and S. albus[22]. To compare different Streptomyces strains for short-chain ketone production, we selected S. coelicolor A3(2) (hereafter referred to S. coelicolor, a close relative of S. lividans) and S. albus J1074 (hereafter referred to S. albus) for experimentation. S. venezuelae was also examined because the parental PKS gene (LipPks1 + TE) was well-expressed in this host after optimizing the TSS as mentioned above. In addition, since the majority of the ethyl and methyl ketone-generating PKSs originates from LipPks1, S. aureofaciens was also included in the experiments described below.

To investigate the impact of using different Streptomyces hosts for short-chain ketone production, we constructed actinophage ΦC31 attP/int vectors carrying an apramycin resistance gene and the genes for mCherry (negative control), N-terminal tail-truncated ethyl ketone PKS, or N-terminal tail-truncated methyl ketone PKS, which can be efficiently transferred from E. coli ET12567/pUZ8002 into various Streptomyces[23]. These genes were placed under control of the gapdh promoter from Eggerthella lenta ($P_{gapdh(EL)}$) that was shown to work in S. coelicolor and S. venezuelae at similar strength (Supplementary Table 1)[24]. The introduced vector is site-specifically integrated into the Streptomyces chromosomes that carry a cognate attB site by site-specific recombination with attP mediated by the integrase encoded in each vector. Apramycin resistant transconjugants were isolated and examined. Correct integration of each construct was verified by the polymerase chain reaction (PCR) using primers listed in Supplementary Table 2 (see also Supplementary Fig. 2 and Supplementary Table 3). Each strain was named by the plasmid used (Supplementary Table 4).

Each of the 12 recombinant strains was initially grown in TSB for 2–4 days and then inoculated into shake flasks containing 30 ml of medium 042 (M042). M042 was developed to maximize production of various secondary metabolites by Streptomyces[25] and gave the best titers among several different media tested

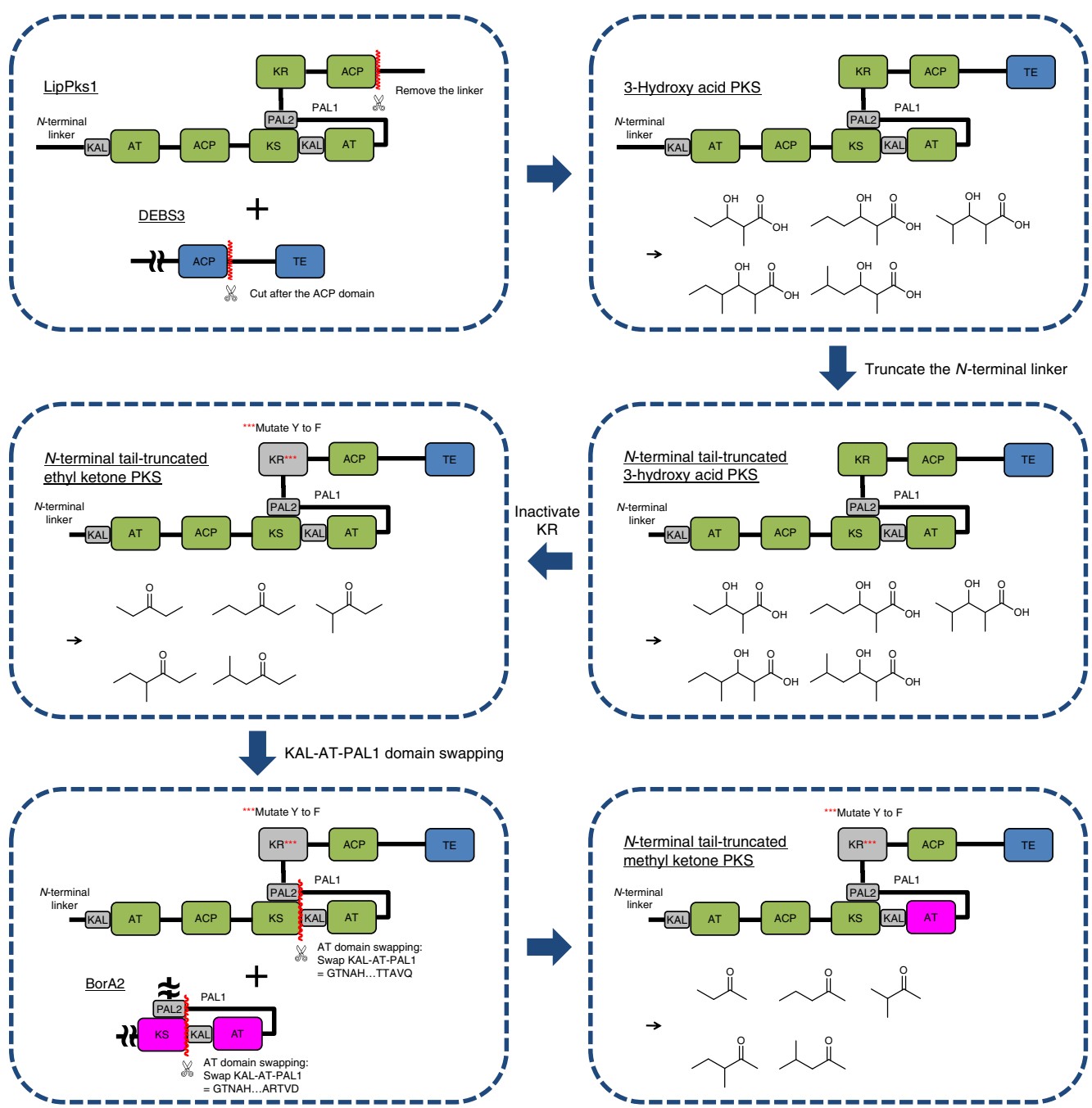

**Fig. 1** Hybrid polyketide synthases engineered to produce short-chain ketones. Short-chain ketone-producing polyketide synthases (PKSs) were created by engineering of LipPks1, a type I modular PKS. Catalytic domains of LipPks1 are shown in green. The non-catalytic domains are shown in gray. Linkers are shown as black line. The TE domain (blue) is derived from the erythromycin PKS. The AT domain shown in magenta is derived from the borrelidin PKS module 1, which is specific for malonyl-CoA. Enzymatic products that can be produced by each PKS were shown in the same box. AT acyltransferase, ACP acyl carrier protein, CoA coenzyme A, KAL KS to AT linker, KR ketoreductase, KS ketosynthetas, PAL Post AT linker, TE thioesterase

(Supplementary Fig. 3 and Supplementary Table 5). Each flask was fitted with a silicon sponge closure, which permits exchange of gasses but reduces evaporation of water and short-chain ketones compared to normal flask caps. After culturing for five days at 30 °C, the culture broth was mixed with an equal volume of methanol and incubated at 50 °C overnight to fully convert 3-keto acids produced by PKSs to the corresponding ketones (Supplementary Fig. 4). After filtration, crude extracts were analyzed by liquid chromatography time-of-flight mass spectrometry (LC-TOF-MS). As shown in Fig. 2a, all of the strains transformed with the gene encoding the ethyl ketone PKS

produced expected 2-methyl-3-pentanone and 4-methyl-3-hexanone, which has never been produced biologically. Among them, *S. albus* produced the highest amounts of the ketones ($106 \pm 6$ mg·l$^{-1}$ of 2-methyl-3-pentanone and $39 \pm 3$ mg·l$^{-1}$ of 4-methyl-3-hexanone). We also tested a *S. albus* strain that encodes *N*-terminal tail-truncated ethyl ketone PKS that harbors S to A mutation in the KR domain. As expected, the S-to-A mutant produced lower amounts of ketones compared to the Y-to-F mutant (Supplementary Fig. 1b). For methyl ketones (Fig. 2b), *S. albus* was the only strain that produced expected 3-methyl-2-butanone ($18 \pm 6$ mg·l$^{-1}$) and 3-methyl-2-pentanone

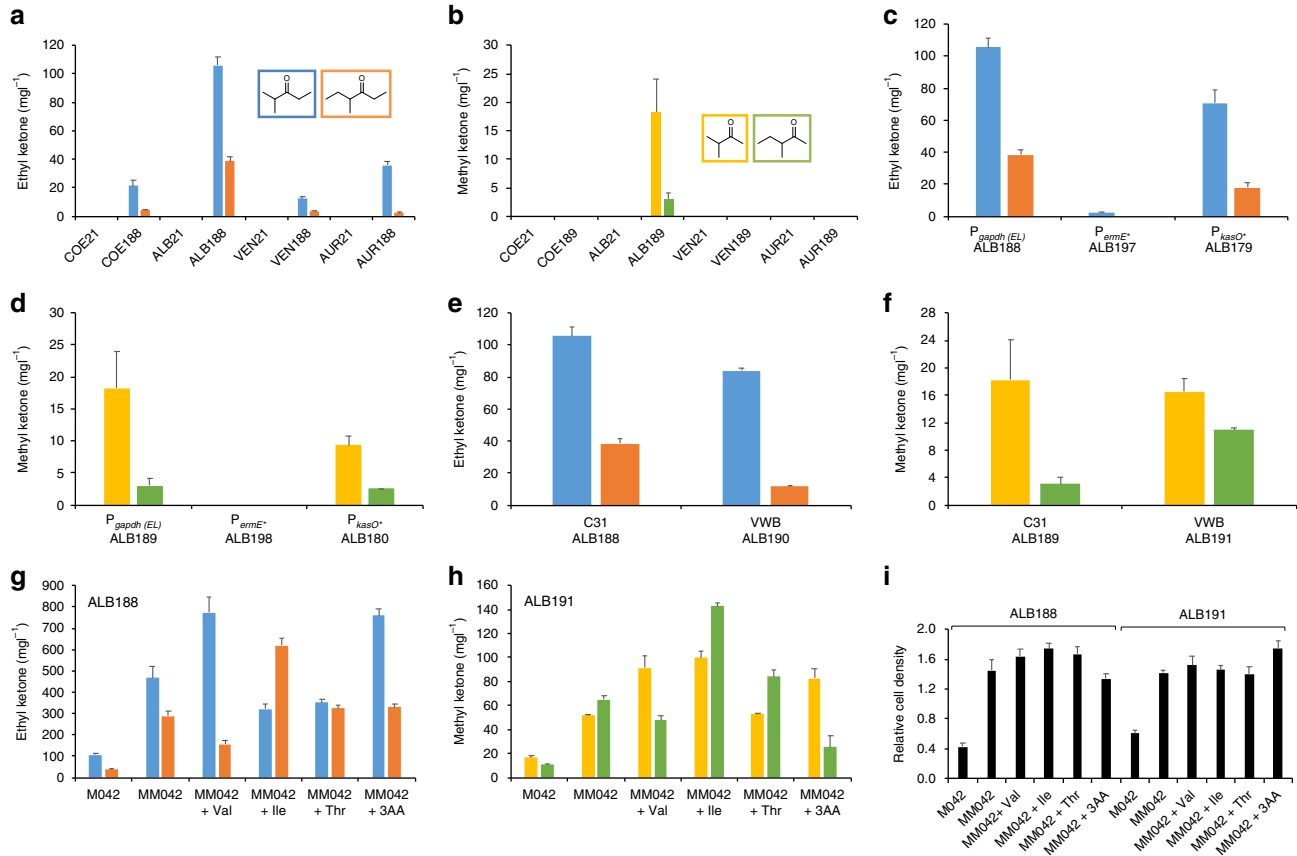

**Fig. 2** Short-chain ketone production in engineered *Streptomyces* strains. Unless otherwise noted, each strain was cultured in Medium 042 (M042) for 5 days at 30 °C and short-chain ketone production was measured by liquid chromatography time-of-flight mass spectrometry. Error bars are the S.D. from three independent experiments. mCherry encoding strains (COE21, ALB21, VEN21, and AUR21) were used as a negative control. **a** Ethyl ketone production from 4 different *Streptomyces* hosts (COE188, ALB188, VEN188, and AUR188). 2-Methyl-3-pentanone and 4-methyl-3-hexanone are shown in blue and orange, respectively. 4-Methyl-3-hexanone was quantified as 5-methyl-3-hexanone equivalent because 4-methyl-3-hexanone is not commercially available. **b** Methyl ketone production from four different *Streptomyces* hosts (COE189, ALB189, VEN189, and AUR189). 3-Methyl-2-butanone and 3-methyl-2-pentanone are shown in yellow and green, respectively. **c** Three different promoters ($P_{gapdh(EL)}$, $P_{ermE*}$, and $P_{kasO*}$) were compared for ethyl ketone production in *S. albus* (ALB188, ALB197, and ALB179). **d** Three different promoters were compared for methyl ketone production in *S. albus* (ALB189, ALB198, and ALB180). **e** Two different genome locations (specific *attB* sites for ΦC31 and VWB) were compared for ethyl ketone production in *S. albus* (ALB188 and ALB190). **f** Two different genome locations were compared for methyl ketone production in *S. albus* (ALB189 and ALB191). **g** Ethyl ketone production by ALB188 in modified medium 042 (MM042) or MM042 supplemented with amino acids (Val, Ile, Thr, or the 3 amino acid mixture). **h** Methyl ketone production by ALB191 in M042, MM042, or MM042 supplemented with amino acids. **i** Relative cell density of ALB188 and ALB191 grown in M042, MM042, or MM042 + amino acids were estimated by the Bradford assay. In MM042 conditions, each strain was cultured for 9 days at 30 °C. ALB *albus,* AUR *aureofaciens,* COE *coelicolor*; VEN *venezuelae*

$(3 \pm 1 \, \text{mg} \cdot \text{l}^{-1})$. Microbial production of the two methyl ketones has not been reported previously. However, the lower titers of methyl ketones were not expected because AT-swapped LipPks1 + TE produced 3-hydroxyhexanoic acid from *n*-butyryl-CoA and malonyl-CoA with an approximately tenfold faster $k_{cat}$ compared to 2-methyl-3-hydroxyhexanoic acid production from *n*-butyryl-CoA and methylmalonyl-CoA by LipPks1 + TE[17]. In fact, the *S. albus* strain that harbors the AT-swapped LipPks1 + TE produced $42 \pm 1 \, \text{mg} \cdot \text{l}^{-1}$ of 2-methyl-3-hydroxypentanoic acid, whereas the *S. albus* strain harboring the original LipPks1 + TE produced 8.8-fold less 2,4-dimethyl-3-hydroxypentanoic acid (Supplementary Fig. 5c). Shorter ketones, in theory, evaporate faster, which, in part, may explain lower methyl ketone titers observed.

**S. albus engineering**. To improve ketone production levels in *S. albus*, we tested different constitutive promoters to drive expression of the hybrid PKS genes (Supplementary Table 1). We employed the ermE* promoter ($P_{ermE*}$), the best characterized promoter used in many *Streptomyces* strains[26], or the recently developed kasO* promoter ($P_{kasO*}$), which showed approximately 50% stronger activity in *S. coelicolor* and *S. venezuelae* than $P_{gapdh(EL)}$[24,27]. The newly constructed *S. albus* strains produced lower amounts of ethyl ketones and methyl ketones compared to the corresponding $P_{gapdh(EL)}$ strains (Fig. 2c,d and Supplementary Table 4). Although we could detect similar levels of short-chain ketones from strains that encode PKS genes under the kasO* promoter, it appears that the strength of ermE* promoter is very low in *S. albus*.

We previously developed *Streptomyces* integration vectors that encode actinophage VWB attP/int or BT1 attP/int, which can be used to insert DNA into specific chromosomal locations of a target genome[23]. Although the frequency of obtaining transconjugants of *S. albus* with a vector containing BT1 attP/int was very low, reasonable numbers of transconjugants were observed using the VWB attP/int system. Using this system, we constructed *S. albus* strains with a gene that encodes an *N*-terminal tail-truncated ethyl

ketone PKS or methyl ketone PKS under control of $P_{gapdh(EL)}$ (Supplementary Table 4). As shown in Fig. 2e, the ethyl ketone production levels were lower than those produced by the corresponding ΦC31 strains. However, the methyl ketone production was slightly higher (Fig. 2f). Although the amount of 3-methyl-2-butanone was similar ($18 \pm 6 \, mg \cdot l^{-1}$ vs. $16 \pm 2 \, mg \cdot l^{-1}$), ca. fourfold higher production was observed for 3-methyl-2-pentanone.

**Short-chain ketone production from plant biomass**. We selected ALB188 and ALB191 to examine short-chain ketone production from plant biomass. These two strains gave the highest titers of ethyl and methyl ketones in M042. M042 contains glucose (1%), glycerol (1%), and corn starch (1%) as major carbon sources. We replaced these carbon sources with corn stover alkaline hydrolysates, which contain glucose and xylose at approximately 7:3 molar ratio. We prepared modified M042 (MM042) using the plant biomass hydrolyate, resulting in final concentrations of glucose and xylose at 6.5% and 2.2%, respectively (Supplementary Table 5).

Although it took approximately 9 days to consume all of the carbon in the MM042 (Supplementary Fig. 6), as shown in Fig. 2g, h (see also Supplementary Figs 7–9), $466 \pm 54 \, mg \cdot l^{-1}$ of 2-methyl-3-pentanone and $289 \pm 21 \, mg \cdot l^{-1}$ of 4-methyl-3-hexanone were produced by ALB188 ($755 \pm 75 \, mg \cdot l^{-1}$ in total), and $52 \pm 1 \, mg \cdot l^{-1}$ of 3-methyl-2-butanone and $65 \pm 3 \, mg \cdot l^{-1}$ of 3-methyl-2-pentanone were produced by ALB191 ($116 \pm 4 \, mg \cdot l^{-1}$ in total). The cell density of each culture was estimated by the Bradford assay (Fig. 2i), and these data suggest that the increased levels of short-chain ketones were mainly achieved by the increased amounts of cell mass generated in media containing plant biomass.

We also tested if we could change the product ratios by changing intracellular concentrations of acyl-CoA substrates. To do this, we supplemented MM042 with valine, isoleucine, or threonine. These amino acids could potentially be converted to isobutyryl-CoA, 2-methylbutyryl-CoA, and propionyl-CoA, respectively, by endogenous enzymes. Propionyl-CoA could be further converted to methylmalonyl-CoA. As expected, when valine was added to the medium, levels of short-chain ketones produced from isobutyryl-CoA (2-methyl-3-pentanone and 3-methyl-2-butanone) increased significantly, whereas higher titers of 4-methyl-3-hexanone and 3-methyl-2-pentanone were observed when isoleucine was added into the medium (Fig. 2g, h). Short-chain ketone production levels were not dramatically affected in the medium containing threonine. When all of the three amino acids were added to the medium, the highest level of ethyl ketone production was observed ($758 \pm 35 \, mg \cdot l^{-1}$ of 2-methyl-3-pentanone and $329 \pm 13 \, mg \cdot l^{-1}$ of 4-methyl-3-hexanone, $1087 \pm 47 \, mg \cdot l^{-1}$ in total, Fig. 2g) with ~4% of maximum theoretical yield. For methyl ketones, the medium containing isoleucine gave the best titer ($99 \pm 6 \, mg \cdot l^{-1}$ of 3-methyl-2-butanone and $143 \pm 3 \, mg \cdot l^{-1}$ of 3-methyl-2-pentanone, $242 \pm 3 \, mg \cdot l^{-1}$ in total, Fig. 2h) with ~1% of maximum theoretical yield. These feeding experiments demonstrate that short-chain ketone production can be controlled by addition of amino acids to the growth medium, although using a strain that overproduces target amino acids would be favorable in an industrial application to reduce fermentation cost. While the kinetic parameters of LipPks1 + TE indicate that other short-chain ketones such as butanone and 4-methyl-2-pentanone could also be produced in vivo from the corresponding starter substrates, propionyl-CoA and isovaleryl-CoA[28], it would be necessary to eliminate production of isobutyryl-CoA and 2-methylbutyryl-CoA, substrates for which the AT has lower $K_m$ values than for propionyl-CoA and isovaleryl-CoA[29].

We also tested whether we could produce short-chain ketones in a bioreactor. Significant evaporation of target ketones prevented the use of the system, however (Supplementary Fig. 10). The reason is still unclear but one possible reason is that higher mass transfer of oxygen in the bioreactor results in more rapid conversion of sugars to 3-keto acids, which could be converted to the corresponding ketones at a faster rate because of more intense aeration and agitation. These conditions would also accelerate stripping of ketones from the fermentation broth, resulting in more rapidly decreasing titers once the ketone removal rate exceeds the production rate.

The use of organic solvent overlay has been shown to be a successful method for accumulating volatile products produced by microbial cultures[30]. To overcome the evaporation issue during fermentation, we selected decane and dodecane as overlays and tested the efficacy by growing ALB188 in medium 042 in the presence of those organic solvents. Unfortunately, we observed lower amounts of ketones. According to the cell density measurements, decane and dodecane may be toxic to *S. albus* (Supplementary Fig. 11).

**Short-chain ketones as gasoline oxygenates**. Ethanol is a renewable oxygenate added to gasoline. Unless it is used in a flex-fuel vehicle, no more than 15% ethanol can be added to the gasoline due to EPA regulation. Because there are few flex-fuel vehicles in the US, there is a limited amount of ethanol needed, which the US has more than enough capacity to produce. Having oxygenates that can be blended into gasoline at greater than 15% would allow more renewable substitutes in gasoline and would lower GHG emissions.

The methyl and ethyl ketones we biologically produced could potentially be used as gasoline replacements or oxygenates in gasoline. Research octane number (RON) and Motor octane number (MON) of eight different short-chain ketones have been reported[14]. Importantly, our engineered PKSs can produce six of them (butanone, 2-pentanone, 3-pentanone, 3-methyl-2-butanone, 3-hexanone, and 4-methyl-2-pentanone). Butanone, 2-pentanone, 2,4-dimethyl-3-pentanone, and cyclopentanone were blended into a model gasoline surrogate, which suggested their ability to increase the RON and MON[14]. Here, we further expanded the analysis by blending ten different short-chain ketones (C3–C7) into an actual gasoline, CARBOB, at 10, 20, and 30 Vol%. CARBOB is a special Reformulated Blendstock for Oxygenate Blending formulae mandated by the state of California (E10 gasoline = CARBOB + Ethanol at 10 Vol%). Our analysis indicates that all short-chain ketones tested, including ketones produced by our engineered *S. albus* strains, increase RON and MON of CARBOB (Fig. 3a, b). Furthermore, most of them behaved better than 1-butanol, a well-known biofuel, although none of them exceeded the performance of ethanol in terms of octane number. However, as shown in Supplementary Table 6, 2-methyl-3-pentanone and 3-methyl-2-butanone have very similar fuel properties as isooctane (octane numbers, energy density, boiling point, melting point, and flash point), and outperform ethanol in terms of energy density, auto ignition temperature, and water solubility. 4-Methyl-3-hexanone and 3-methyl-2-pentanone (which were also produced from the *S. albus* strains) were not included in the blending study because they were either not commercially available or too expensive for the test.

**Discussion**

Microbial production of fuels and commodity chemicals has been performed primarily using natural enzymes, which inherently limits the types of molecules that can be produced. Engineering natural enzymes (single to a few amino acid mutations) has been

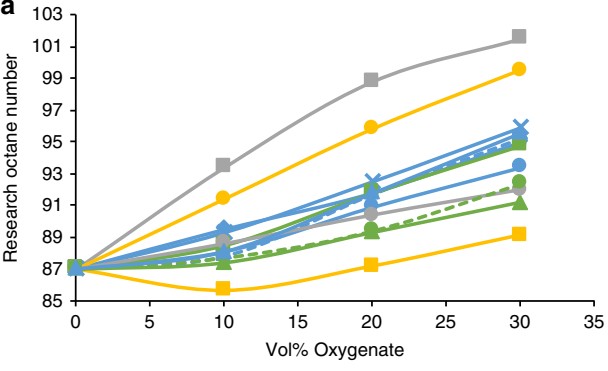

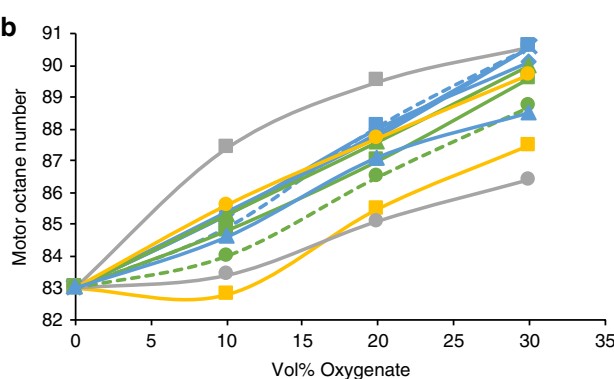

**Fig. 3** Octane numbers of short-chain ketones blended into a model gasoline. Three ethyl ketones (green), five methyl ketones (blue), other ketones (yellow), ethanol (gray square), and 1-butanol (gray circle) were added into CARBOB at 10, 20, and 30 Vol%. **a, b** Research octane numbers (**a**) and Motor octane numbers (**b**) were measured by ASTM methods D2699 and D2700, respectively. Short-chain ketones produced by engineered *S. albus* strains were shown as a dashed line. 2-methyl-3-pentanone (green circle), 3-hexanone (green triangle), 3-pentanone (green square), 4-methyl-2-pentanone (blue circle), acetone (blue triangle), 3-methyl-2-butanone (blue square), 2-pentanone (blue diamond), butanone (blue cross), cyclopentanone (yellow circle), 2,4-dimethyl-3-pentanone (yellow square). CARBOB Reformulated Blend-stock for Oxygenate Blending formula mandated by the state of California

shown to be effective in broadening the substrate or product range or changing the enzyme kinetics, but generally the titers of the engineered enzymes are significantly reduced relative to the natural enzymes; to our knowledge, the highest titer reported for biogasoline production (e.g. short-chain alkanes or alcohols) was $728\,\mathrm{mg\cdot l^{-1}}$ when using these mutated enzymes[9,11,31,32]. Type I modular PKSs have the potential to produce a very broad range of molecules by combining multiple, simple individual steps (catalyzed by protein domains) in an understandable way. By isolating and characterizing each step and reconstituting them in various combinations, these systems have been shown to produce complex chemical structures not observed in nature. Unfortunately, engineered PKSs generally have significantly reduced kinetics compared to their wild-type counterparts and, as a result, produce low titers of the desired product. In the present work, we demonstrated efficient production of potential fuels using a hybrid PKS at a titer exceeding $1\,\mathrm{g\cdot l^{-1}}$. It should be noted that actual ketone production titers should be higher because 3-keto acids are chemically unstable and gradually converted into the corresponding ketones during fermentation (Supplementary Fig. 4b), which evaporate from cultures at very fast rates (Supplementary Fig. 11a). To our knowledge, this is the highest titer of products generated with any engineered PKSs and the first report

of engineered PKSs used to produce biogasolines. The short-chain ketones produced by the engineered PKS could also be used as solvents, flavors, and fragrances. In addition, our ketone PKSs produced the expected products with no observable off-target products. As shown in Supplementary Figs 7–9, we observed expected ketone products only (no peak shoulders and split peaks). These non-natural hybrid enzymes were created by physically combining different parts of wild-type PKSs using a previously developed and recently reported PKS engineering strategy to replace the native extension AT with a non-native AT with highly conserved amino acid sequences, GTNAHVILE and LPTY(A/P)FQ(H/R)xRYWL[17]. These domain boundaries are very well conserved in type I modular PKSs[33]. Interestingly, very similar sequences can also be found in type I iterative PKSs, which could also be used to change substrate specificity without disturbing protein quaternary structures. We also employed an alternative TSS for LipPks1 to redesign ketone PKSs. Gene-finding tools perform well in low GC content genomes but are much less accurate for high GC genomes such as *Streptomyces* genomes[34]. In fact, we found a TSS that greatly outperforms the originally annotated start codon in heterologous expression of LipPks1[20]. Because most bioactive PKS products originate from *Strepromyces*, more accurate gene-finding tools would facilitate heterologous expression of the PKSs. We further demonstrated that organisms engineered with these PKSs could utilize cellulosic biomass to produce the target molecules. Together, our work suggests that precisely engineered PKSs could be used to produce a broad range of hydrocarbons that we currently derive from fossil resources plus many molecules that we could never produce from fossil resources in an efficient manner from renewable carbon sources.

## Methods

**Chemicals**. All chemicals were purchased from Sigma-Aldrich (United States) unless otherwise described.

**Plasmids and strains**. Plasmids and strains used in this study are listed in Supplementary Table 4. The plasmids and strains have been deposited in the public version of JBEI registry (http://public-registry.jbei.org) and are physically available from the corresponding author upon request.

**Plasmid construction**. Briefly, recognition sequence of SpeI was added to *Streptomyces* integration vector (apramycin[R], VWB)[23] using primers listed in Supplementary Table 7 to replace the integrase with SphI and SpeI and the promoter with SpeI and NdeI (Supplementary Table 7). Using these sites, p21 was constructed. p33 was constructed from p21 by replacing the mCherry gene with a PKS gene from pSY074[17] using NdeI and EcoRI. Similarly, P37 was constructed using a PKS gene generated with NdeI and EcoRI from pSY075[17]. pSY172 encodes *N*-terminal tail-truncated LipPks1 + TE[20]. The PKS gene was cut with NdeI and EcoRI and inserted into p21 (pre-pSY186). To inactivate the KR domain, the KpnI-NheI fragment from pSY087 that contains LipPks1 AT-KR-ACP tri-domain, in which the catalytic serine in the KR was mutated to alanine[17], was used to replace the corresponding region in pre-pSY186 to construct pSY186. The KpnI-NheI fragment from pSY074 was subcloned into a smaller vector, and the catalytic tyrosine in the KR was mutated to phenylalanine by site-directed mutagenesis using primers listed in Supplementary Table 7. The KpnI-NheI fragment was used to replace the corresponding region in pSY186 to construct pSY188. The NdeI-EcoRI fragment from pSY188 was used to construct all of the other plasmids that harbor the gene encoding the ethyl ketone PKS. Similarly, the KpnI-NheI fragment from pSY075 was subcloned into a smaller vector and the catalytic tyrosine in the KR was mutated to phenylalanine by site-directed mutagenesis using the same primers. The KpnI-NheI fragment was used to replace the corresponding region in pSY188 to construct pSY189. NdeI-EcoRI fragment from pSY189 was used to construct all of the other plasmids that encode methyl ketone PKS (see also Supplementary Fig. 12).

**Conjugal transfer of vectors to *Streptomyces***. Conjugal transfer of integration vectors to *Streptomyces* hosts was conducted using the method described in Practical Streptomyces Genetics with slight modifications[35]. *E. coli* ET12567/pUZ8002 was transformed with a desired plasmid employing selection on LB agar containing apramycin ($50\,\mathrm{mg\cdot l^{-1}}$). A single colony was inoculated into 5 ml of LB containing kanamycin ($25\,\mathrm{mg\cdot l^{-1}}$), chloramphenicol ($15\,\mathrm{mg\cdot l^{-1}}$), and apramycin

(25 mg · l⁻¹) and grown at 37 °C. The overnight culture was used to seed 10 ml of LB containing the same antibiotics (inoculum size was 5 Vol%), which was grown at 37 °C to an $OD_{600}$ of 0.4–0.6, which usually takes 4–6 h. The *E. coli* cells were pelleted by centrifugation at $3000 \times g$ for 5 min, washed twice with 10 ml of LB, and resuspended in 1 ml of LB. Fresh *Streptmyces* spores were collected from a glycerol-arginine plate[23] (*S. venezuelae*) or Mannitol Soy (MS) agar plate (*S. albus, S. aureofaciens,* and *S. coelicolor*) with 5 ml of 2x YT and incubated at 50 °C for 10 min. MS agar was prepared by adding 20 g of agar, 20 g of mannitol, and 20 g of soybean flour to 1 l of tap water and autoclaved. After autoclaving and cooling, magnesium chloride was added at 10 mM. The spores (500 µl) and the *E. coli* cells (500 µl) were mixed and briefly spun down at $3000 \times g$ for 2 min. Most of the supernatant was poured off and spread on MS agar, and incubated at 23 °C (*S. venezuelae*) or 30 °C (*S. albus, S. aureofaciens, S. coelicolor*) for 16–20 h. After addition of 1 ml of water containing nalidixic acid (0.5 g · l⁻¹) and apramycin (1.25 g · l⁻¹), the plate was further incubated for 3–5 days to permit sporulation. A single colony was inoculated into 3 mL of Tryptic Soy Broth (TSB) non-animal origin (EMD Millipore, United States) containing nalidixic acid (25 mg · l⁻¹) and apra-mycin (25 mg · l⁻¹). Following 2–4 days incubation at 30 °C, 100 µl of the culture was spread onto glycerol-arginine agar (*S. venezuelae*) or MS agar (*S. albus, S. aureofaciens,* and *S. coelicolor*). The plate was incubated for 4–7 days at 30 °C. Spores were collected from the plate by adding 5 mL of water. The spore suspension was filtered using a syringe containing a piece of cotton wool. The resulting suspension was mixed with equal volume of glycerol.

**Genomic PCR.** Gene integration was assessed by amplifying the corresponding DNA regions using primers listed in Supplementary Table 2. PCR was carried out in 50 µl total reaction volumes, each containing spores, 0.5 µM of each primer, 10 µl of 5x HF buffer (Thermo Scientific, United States), 200 µM dNTPSs, 1 M Betaine, and 0.5 µl of Phusion (Thermo Scientific, United States). The reaction mixture was heated to 98 °C for 2 min, followed by 35 cycles, each consisting of 5 sec dena-turation at 98 °C, 10 sec annealing at 61 °C, 1.5 min extension at 72 °C, and a final 7 min extension at 72 °C. The sequence of each PCR fragment containing the promoter was also confirmed.

**Microbial ketone production.** Engineered *Streptomyces* spores (50 µl) were grown in 3 ml of TSB containing nalidixic acid (25 mg · l⁻¹) and apramycin (25 mg · l⁻¹) for 2–4 days at 30°. *S. coelicolor, S albus,* and *S. aureofaciens* grow more slowly than *S. venezuelae* and usually took 3–4 days to have enough growth. 0.5 ml of overnight culture was transferred to a 250 ml non-baffled flask containing 30 ml of 042 medium[25], nalidixic acid (25 mg · l⁻¹) and apramycin (25 mg · l⁻¹) fitted with a silicon sponge cap (https://www.sigmaaldrich.com/catalog/product/sigma/c1046?lang=en®ion=US). The culture was grown for 5 days at 30 °C at 200 rpm on a Kuhner ISG-1-W shaker (Kuhner, United States). Furthermore, 300 µl of super-natant was collected from the culture, mixed with 300 µl of methanol, and then incubated at 50 °C overnight (the tube was covered with parafilm). The resulting solution was cooled to 4 °C, centrifuged at $10000 \times g$ for 2 min, and then filtered using an Amicon Ultra Centrifugal filter, 3 kDa Ultracel, 0.5 mL device at $10000 \times g$ for 10 min (EMD Millipore, United States). The flow-through was analyzed by liquid chromatography time-of-flight mass spectrometry (LC-TOF).

The same experimental procedures were used to produce and extract ketones from engineered *S. albus* strains grown in modified Medium 042 except that the culture was grown for 9 days to fully consume sugars derived from plant biomass.

In a bioreactor, engineered *S. albus* spores were grown in 3 ml of TSB containing nalidixic acid (25 mg · l⁻¹) and apramycin (25 mg · l⁻¹) for 2–4 days at 30 °C. Seed culture grown on TSB was used to inoculate the second seed with 50 ml of Medium 042 (inoculum size was 8 Vol%). The second seed was grown in shake flasks for overnight at 30 °C. 20 ml of overnight culture was transferred to a 2 l bioreactor containing 700 ml of 042 medium, nalidixic acid (25 mg · l⁻¹) and apramycin (25 mg · l⁻¹). Fermentation conditions were controlled to achieve a temperature of 30 °C, to maintain pH 7.2, and to dissolve oxygen at 40% of dissolved oxygen saturation. Dissolved oxygen was controlled by setting up a cascade on agitation and air flow. The minimum and maximum cascade range for agitation was 200 and 400 rpm respectively; for aeration the range was 0.5 and 1.0 LPM. The cascade controlled the dissolved oxygen by first changing the agitation speed and then air flow rate. The pH was controlled at 7.2 by addition of 2 M NaOH. The foaming issue was controlled by addition of 5 Vol% of Antifoam 204.

**Streptomyces cell growth measurement.** Growth of *Streptomyces* cells was measured using the Bradford assay. Upon harvest of shake flasks 1 ml of the culture was spun down at $10,000 \times g$ for 5 min and supernatant was decanted. Cells were then washed with deionized water once and resuspended in deionized water to final volume of 1 ml. Furthermore, 20 µl of washed cells were mixed with 1 ml of Quick Start Bradford 1x Dye Reagent (BIO-RAD, United States) and absorbance of the mixture was measured at 595 nm immediately at room temperature.

**Ketone production analysis.** LC separation of ketones was conducted on a Kinetex XB-C18 reversed phase column (100 mm length, 2.1 mm internal dia-meter, 2.6 µm particle size; Phenomenex, United States) using an Agilent 1200 Rapid Resolution LC system (Agilent Technologies, United States). The mobile

phase was composed of water (solvent A) and methanol (solvent B). Ketones were each separated via the following gradient: increased from 35 to 90.4% B in 4.4 min, held at 90.4% B for 2.2 min, decreased from 90.4 to 35% B in 0.2 min, and held at 35% B for an additional 2.7 min. The flow rate was held at 0.18 ml · min⁻¹ for 4.4 min, increased from 0.18 to 0.4 ml · min⁻¹ in 0.2 min, and held at 0.4 ml · min⁻¹ for an additional 4.9 min. The total LC run time was 9.5 min. The column com-partment and autosampler temperatures were set to 50°C and 6°C, respectively. Samples were injected into the LC column at a volume of 10 µl. The Agilent 1200 Rapid Resolution LC system was coupled to an Agilent 6210 TOF (Agilent Technologies, United States). Nitrogen gas was used as both the nebulizing and drying gas to facilitate the production of gas-phase ions. The drying and nebulizing gases were set to 10 l · min⁻¹ and 25 l · bin⁻², respectively, and a drying gas tem-perature of 325°C was used throughout. Atmospheric pressure chemical ionization was conducted in the positive-ion mode with capillary and fragmentor voltages of 3.5 kV and 100 V, respectively. The skimmer, OCT1 RF, and corona needle were set to 50 V, 170 V, and 4 µA, respectively. The vaporizer was set to 350°C. Ketones were detected via [M + H]+ ions: $m/z = 87.08044$; $m/z = 101.09609$; $m/z = 115.11174$. The analysis was performed using an $m/z$ range of 66 to 166. Data acquisition and processing were performed using MassHunter software (Agilent Technologies, United States). For 4-methyl-3-hexanone, because the authentic standard is not commercially available, the isomer, 5-methyl-3-hexanone was used to quantify the product.

**Biomass pretreatment and saccharification.** A mixture containing 15% corn stover biomass (7% w/w moisture), 1.5% NaOH, and 83.5% water was pretreated by autoclaving at 121 °C for 1 h. Following pretreatment, the biomass was wrapped in cheesecloth and dried in a laundry centrifuge to approximately 30% w/w solids. The supernatant was discarded, and biomass was re-suspended and soaked in deionized water overnight with pH adjusted to 5.0. The preparation was then centrifuged a second time to remove excess salt and moisture. Pretreated biomass was saccharified in 2-L LR-2.ST IKA reactors (IKA, United States) using com-mercially available enzymes CTec2 and HTec2 (Novozymes, Denmark). Enzymes with following loadings were added to the reactor: 64 mg CTec2 · g⁻¹ dry biomass and 8 mg HTec2 · g⁻¹ dry biomass. Enzymatic saccharification was performed at 50 °C with pH in the range of 4.5 to 5.5 for 96 h. Upon completion of the sac-charification reactions, the unhydrolyzed biomass was separated from the hydro-lysate by centrifugation at $4000 \times g$ for 30 min. The hydrolysate was filtered with 0.7 µm and then 0.45 µm filter papers to separate any remaining particles and finally sterilized by passing through 0.2-µm filters and stored at 4 °C for further use. The final hydrolysate contained 86.5 g · l⁻¹ of free glucose and 38.1 g · l⁻¹ of free xylose.

**Sugar consumption analysis.** Glucose and xylose concentrations in the aqueous portion of fermentation samples were quantified by high pressure liquid chro-matography (HPLC). This method is suitable for detection and quantification of individual sugars. The apparatus used was Ultimate 3000 HPLC system® (Thermo Scientific, United States) equipped with a Shodex Refractive Index® detector (Shoko Scientific Co., Ltd., Japan). The carbohydrates were separated on an Aminex HPX-87H (Bio-Rad, United States) column with 9 µm particle size and 300 mm × 7.8 mm dimensions. A standard cartridge holder was used to protect the column. The mobile phase was 0.01 N sulfuric acid with a flow rate of 0.6 ml · min⁻¹, the oven temperature was 65 °C, and the analysis time was 30 min. Integration and analysis of samples was performed using Dionex Chromeleon® software (Thermo Scientific, United States). Identification of monosaccharide content was determined relative to known standards. Limits of detection for glu-cose and xylose were .80 mg · l⁻¹.

**Octane number analysis.** A special Reformulated Blendstock for Oxygenate Blending formulae mandated by the state of California (CARBOB) was used as a base gasoline. Each ketone (or alcohol) purchased from Sigma-Aldrich (United States) or TCI (Japan) was blended at 10, 20, or 30 Vol% into 500 ml of CARBOB. Research octane number and Motor octane number for ketone or alcohol-blended gasolines were determined using ASTM methods D2699 and D2700, respectively, by Intertek (United States).

**Auto ignition temperature analysis.** 2-Methyl-3-pentanone and 3-methyl-2-butanone purchased from Sigma-Aldrich (United States) or TCI (Japan) were analyzed using ASTM methods E659 by Alcor Petrolab (United States).

**GC-FID analysis.** After the decane or the dodecane layers were sampled, they were analysed by GC-FID (Thermo Focus with FID) equipped with a DB-5 column (30 m, 0.32 mm ID, 0.5 µm film), using a the corresponding standard curve using the following conditions: inlet at 250 °C, FID at 250 °C, 50 kPa constant pressure; oven: 35 °C, 5 °C min⁻¹ to 60 °C, 80 °C min⁻¹ to 250 °C, and the hold for 5 min.

## Data availability

The DNA sequences of plasmids and strains used in this study have been deposited in the public version of JBEI registry (http://public-registry.jbei.org). Please refer to Supplementary Table 4 for accession codes.

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

## Acknowledgements

This work was funded by the Joint BioEnergy Institute, which is funded by the U.S. Department of Energy (DOE), Office of Science, Office of Biological and Environmental Research, and the Co-Optimization of Fuels & Engines project sponsored by the U.S. DOE Office of Energy Efficiency and Renewable Energy, Bioenergy Technologies and Vehicle Technologies Offices, under Contract DEAC02-05CH11231 between DOE and Lawrence Berkeley National Laboratory. This work was also funded by the National Science Foundation under award MCB-1442724.

## Author contributions

S.Y., T.F. and R.J. constructed plasmids and engineered Streptomyces strains. S.Y., M.M., F.M., E.S. and D.T conducted fermentation. S.Y., V.T.B. and E.E.K.B conducted metabolite analysis by LC-TOF. S.Y., A.G., J.M.G. and R.W.D. conducted octane number analysis. S.Y., T.R.P., B.A.S., L.K. and J.D.K. were responsible for experimental design. All authors contributed to the preparation of the manuscript.

## Additional information

**Competing interests:** J.D.K. has a financial interest in Amyris, Lygos, Demetrix, Constructive Biology, Napigen, and Maple Bio. L.K. has a financial interest in Lygos. All other authors declare no competing interests.

