## [Peer Review File · Nature Communications]

Reviewers' comments:

Reviewer #1 (Remarks to the Author):

The authors of this manuscript titled "Microbial production of short-chain ketones using hybrid polyketide synthases" employed engineered *Streptomyces* species expressing hybrid polyketide synthases (PKSs) to produce a series of short-chain ketones. Rational engineering strategies including selection of PKS with substrate promiscuity, replacement of acyl transferase (AT) domain with a non-native AT domain, and use of alternative translation start site to increase the PKS expression, allowed high-level production of short-chain (C5 – C7) ketones using recombinant *Streptomyces albus* strains. The authors further demonstrated the production of the short-chain ketones using plant biomass as a sole carbon source and successful use of the short-chain ketones as gasoline additives. This is a well-written manuscript with a solid set of experiments. This manuscript can be further improved for publication after revision.

Major comments

1. Results, Rational PKS engineering, paragraph 2 & Supplementary Fig. 6: Please provide the titers of the short-chain ketones produced with the unmodified KR domain. In addition, it seems like both the S1887A and Y1900F mutations give positive effects on the production of the short-chain ketones. Hence, the combination of the two mutations is likely better for the production of the ketones unless the Y1900F completely ablated the function of KR. Please provide the titer of the short-chain ketones produced with the mutant KR domain harboring both the S1887A and Y1900F mutations.
2. Results, *Streptomyces* host selection, paragraph 1: Please describe the reason why the authors chose *S. coelicolor* A3(2) and *S. albus* J1074 as the host strains for the production of the short-chain ketones
3. According to the description in the section 'Streptomyces host selection' in the Results, keto acids produced by the recombinant bacteria were chemically converted to the corresponding ketones. This should not be described as a bioproduction of ketones but as a bioproduction of keto acids with subsequent chemical conversion to ketones. It is suggested to explicitly mention that chemical conversion was thoroughly utilized, either in the manuscript or in the methods section.
4. Results, *Streptomyces albus* engineering, paragraph 2: Since the ϕ C31r attP/int, VWB attP/int, and BT1 attP/int systems have different integration sites, introduction of three copies of the engineered PKS genes into the three different genomic loci is expected to enhance the production of the ketones. Please include this engineering strategy in the manuscript and describe the enhanced result.
5. Results, Short-chain ketone production from plant biomass: Use of condensers or trapping of the evaporated ketones using organic solvents is expected to overcome the troubles during the fermentation in the bioreactor. Please apply such process engineering strategies to prevent the loss of the ketones by evaporation and accurately examine the capacity of the engineered strains in the bioreactors.
6. Results, Short-chain ketone production from plant biomass: In this section, the authors supplemented amino acids for the production of different ketones. However, it should be discussed whether feeding expensive amino acids are economically feasible. Otherwise, microbial strains capable of producing these amino acids should be utilized as the base strains.
7. Results, Short-chain ketones as gasoline oxygenates: Were the ketones used for the examination extracted from the cell culture? Please clearly mention how the ketones from the cell culture were extracted, and the percentages of the incorporated ketones from the cell culture.

Minor comments

1. Introduction, paragraph 1: Please include proper citation of the report
2. Results, Rational PKS engineering, paragraph 1: Fig. 1 \diamond Fig. 1a
3. Results, Rational PKS engineering, paragraph 2: Please cite 'the recently reported PKS

engineering strategy' in the first sentence

4. Results, Rational PKS engineering, paragraph 2: Supplementary Fig. 6 is mentioned before Supplementary Fig. 1 – 5. The same applies to the supplementary tables. Please reorder the supplementary materials so that the first item is mentioned first in the manuscript.
5. Results, Rational PKS engineering, paragraph 3, 2nd sentence: Please cite appropriate reference supporting this sentence.
6. Results, Rational PKS engineering, paragraph 1, 2nd last sentence: Please cite appropriate reference supporting this sentence.
7. Results, Rational PKS engineering, paragraph 1, 2nd last sentence: Please cite appropriate reference describing the actinophage ϕ C31r attP sites and the corresponding integrase.
8. Results, Rational PKS engineering, paragraph 3: 8.8-fold instead of 9-fold
9. Fig. 1a: Please redesign the 'rational PKS engineering' part to display the strategy more clearly. It is suggested to explicitly describe the before and after constructs of the engineered PKS.
10. Methods, Ketone production analysis: It is mentioned that since the authentic standard for 4-methyl-3-hexanone is not commercially available, 5-methyl-3-hexanone was used to quantify 4-methyl-3-hexanone. However, since the data obtained through this procedure might not be correct, the authors should also mention in the manuscript about this calculation method, such as '4-methyl-3-hexanone was quantified as 5-methyl-3-hexanone equivalent.'

Reviewer #2 (Remarks to the Author):

I was very impressed with the comprehensiveness and excellent quality of the work. This study nicely demonstrates how PKS engineering strategies, in combination with host, promoter and media optimization can significantly improve titers. The work establishes a strong platform for future metabolic and enzyme engineering involving modular PKSs towards the industrial manufacture of a diverse class of hydrocarbons. Nevertheless, I had some specific questions.

1. I am a bit puzzled by the motivation that has been implicitly described in the manuscript for pursuing the synthesis of ketones. Since this has not been explicitly stated, the authors are running the risk of readers filling in the blanks themselves and this could detract from the impact of the work. My own interpretation of the reason that the authors are advocating microbial synthesis of short-chain ketones is that these molecules are presently manufactured from petroleum and that a good bio-based alternative could prove to be sufficient to meet their current demand. The authors also cite an ICIS report that discusses non-fuel uses of short-chain ketones. However, they then change their focus to using these molecules as fuel additives. Here too, the reasoning is a bit suspect (or incomplete, to be precise). While the octane number is an important criterion that is used to assess the performance of a fuel, it is not the only criterion. What about isothermal compressibility and auto-ignition temperature, which, according to fuel researchers, are equally or arguably more important than octane numbers. Why weren't these properties estimated? Additionally, since ethanol outperforms the methyl ketones in regard to octane numbers, it would be appropriate to discuss the 'blending' criteria that disqualify ethanol, and how the methyl ketones in question measure up. Physical reasons beyond octane numbers underpin EPA's 15% blending criteria for ethanol. What if blending of methyl ketones is similarly limited by physical incompatibilities?

2. Figure 1 is very similar to a figure from a previous study reported by the same authors. In fact, the only difference between the current figure and its previous version is the inclusion of a small PKS engineering diagram that is not very informative. I am not sure that the editorial policy for Nature Communications is but I am assuming that permission would be required to re-use part of a copyrighted work.

3. How generalizable is the PKS engineering described by the authors to modular PKS that have been sourced from other strains such as *Mycobacterium* spp.? The lack of generalizability of PKS

engineering across PKS, as well as hosts has hitherto been a significant hurdle for the field.

4. I was very impressed by the results of the three PKS engineering methods detailed in the paper. The authors build a very compelling case for each method. However, these methods were tested under very different conditions and in a very different system from the system described in the current study. As a consequence, what the reader is being presented is actually the combined effect of many engineering strategies at once, along with improved host selection. If one were to perform a principal component analysis, what would the authors expect to see? Alternatively, if one has a hybrid PKS with known activity in *E. coli*, how much improvement can they hope to expect when the PKS is expressed in *S. albus*? An *in vitro* assay, or at least an assay in an *E. coli* host comparing improvements for all new engineering method combinations would be ideal.

5. The difference between LipPKS1+TE and the AT-swapped version is shown in Fig. S2-c, which is helpful. How much of the productivity enhancement comes from a less bulky group (H vs. methyl) near the reactive carbon?

6. It would be appropriate to see primers for engineering work carried out in this study (particularly, for the site modification to the ketoreductase) in the supplementary information.

7. The description of the culturing conditions is quite vague and unhelpful in its current form. For instance, on page 4, the authors state, "Each of the 12 recombinant strains was initially grown in TSB for a few days and then inoculated..." Could the authors be more precise? This information is critical for assessing productivity of the strains. In fact, since the authors are targeting fuel applications for the ketones, productivity is arguably the only metric that should be used to assess performance of the strains.

8. On page 4, the authors state, "M042 was developed to maximize production of various secondary metabolites by *Streptomyces* and gave the best titers among several different media tested (data not shown)." Could this data be provided in the supplementary information? Was this data produced in the current study or a previous study? It would be helpful to see the relative performance, especially since the only difference between the two versions of M042 reported in the study is the replacement of sugars with plant biomass hydrolysates.

9. Do *Streptomyces* spp. hosts naturally have an attB site or does it need to be introduced? If doing so was part of the work for this study, what locations were chosen and what primers/DNA was used to achieve this goal?

10. Could the authors provide some LC-MS chromatograms and spectra in the supplementary information? How precise are the engineered PKS? Is there any by-product formation?

11. Could the authors provide more information about genetic engineering of *S. albus* in the form of protocols (if they were developed during the course of the current study) or citations?

12. In figure S5, which shows ketone production levels in a bioreactor, the maxima is achieved within the first 50 hours. However, in the biomass tests, sugar consumption occurs over 9 days. What is the fractional yield of the ketone from sugars and biomass, respectively? This is of particular interest for these strains' and PKSs' intended roles as providers of renewable energy and chemicals.

Reviewer #3 (Remarks to the Author):

For this review, I prefer to send an anonymized report.

This work demonstrated a route to produce various short-chain (C5-C7) ketones using hybrid PKSs

in *Streptomyces*, and the titers and the following engine tests seem to have great potential for application.

However, the concept has been proofed by previous report (ACS Synth Biol, 2017 6, 139-147) . The Fig 1b and c in the current work is nearly the same as Fig. 4A in the previous article. If the novelty of this work is the improved yield, the authors should clearly charify the novalty of the strategy employed and the detailed steps. Therefore, in my opinion, the design of this work seems lack of enough novelty required by Nat. Commun. But the results are impressive, so I suggest it be submitted to more specific journal.

Other comments:

(1) Too many background information in the first part of results "Rational PKS engineering", Moving this words to Introduction should be better.

(2) In the test, Supplementary Fig. 6 appears first. The authors should order the figures in sequence.

(3) The authors only test three promoters in this work, should other promoters be better? To coordinate the biosynthesis of short-chain ketones and other physiological events within the hosts, fine-tuning the expression using a panel of promoters with different profiles of time and strength, I think, will be helpful.

(4) For rational engineering, two chromosomal locations were tested, it seems a little subjectivity. Might other chromosomal loci be better?

(5) When using plant biomass, the utilization levels and productivities of the engineered strains were the more concerned data. However, the author did not provide the information.

Response to Reviewers:

Reviewer #1 (Remarks to the Author):

The authors of this manuscript titled “Microbial production of short-chain ketones using hybrid polyketide synthases” employed engineered Streptomyces species expressing hybrid polyketide synthases (PKSs) to produce a series of short-chain ketones. Rational engineering strategies including selection of PKS with substrate promiscuity, replacement of acyl transferase (AT) domain with a non-native AT domain, and use of alternative translation start site to increase the PKS expression, allowed high-level production of short-chain (C5 – C7) ketones using recombinant Streptomyces albus strains. The authors further demonstrated the production of the short-chain ketones using plant biomass as a sole carbon source and successful use of the short-chain ketones as gasoline additives. This is a well-written manuscript with a solid set of experiments. This manuscript can be further improved for publication after revision.

Reviewer #1 major comment 1:

1. Results, Rational PKS engineering, paragraph 2 & Supplementary Fig. 6: Please provide the titers of the short-chain ketones produced with the unmodified KR domain. In addition, it seems like both the S1887A and Y1900F mutations give positive effects on the production of the short-chain ketones. Hence, the combination of the two mutations is likely better for the production of the ketones unless the Y1900F completely ablated the function of KR. Please provide the titer of the short-chain ketones produced with the mutant KR domain harboring both the S1887A and Y1900F mutations.

Our response to reviewer #1 major comment 1:

In our previous paper (ACS Synth Biol. 6, 139-147, 2017), we demonstrated that LipPks1+TE (the KR is unmodified) was not capable of producing the corresponding ketones *in vitro*. This was expected as a 3-keto group is required for decarboxylation of the acid. Inactivation of the KR domain would yield the 3-keto-containing compound. The S to A mutation in the KR active site was necessary to produce the ketones. Both ALB186 (S1887 to A) and ALB188 (Y1900 to F) (Fig. S1b) encode an *N*-terminal tail-truncated LipPks1+TE in which the KR domain is inactivated. As the reviewer pointed out, we do not have the data from a control strain that encodes *N*-terminal tail-truncated LipPks1+TE but with an active KR domain. Twice we attempted to construct *N*-terminal tail-truncated LipPks1+TE by restoring the mutations that inactivated the KR. While we were able to generate the appropriate constructs, we were not able to transform them into the strain. We suspect that the corresponding products, 3-hydroxy acids, may be toxic to the cells at the high titers that our very effective *N*-terminally truncated PKS produces. Instead, we tested ketone production from ALB33, which encodes the original LipPks1+TE (the *N*-terminal tail is not truncated). No ketone production was observed from the strain. In Fig. S1b, we tested if the Y to F mutation has a positive effect. The Y is the critical residue for the catalysis although the role of the S could be replaced by other amino acid residues based on the mechanisms shown in Fig. S1a. In theory, the double mutant does not provide the improved activity. Having more mutations may also disturb the protein folding.

Reviewer #1 major comment 2:

2. Results, *Streptomyces* host selection, paragraph 1: Please describe the reason why the authors chose *S. coelicolor* A3(2) and *S. albus* J1074 as the host strains for the production of the short-chain ketones.

Our response to reviewer #1 major comment 2:

We discussed this point in the main text (see Line 111-131).

Reviewer #1 major comment 3:

3. According to the description in the section '*Streptomyces* host selection' in the Results, keto acids produced by the recombinant bacteria were chemically converted to the corresponding ketones. This should not be described as a bioproduction of ketones but as a bioproduction of keto acids with subsequent chemical conversion to ketones. It is suggested to explicitly mention that chemical conversion was thoroughly utilized, either in the manuscript or in the methods section.

Our response to reviewer #1 major comment 3:

This is an excellent point, and we have worked to clarify it in the text. As shown in Fig. S4b, conversion of 3-keto acids to ketones occur even at 23 °C, which is lower than the culture temperature (30 °C). We incubated the resulting supernatants at 50 °C overnight just to accelerate ketone formation and complete the reaction within a day for the subsequent LC-MS measurement. Spontaneous decarboxylation of 3-keto acids has been utilized to produce medium-chain methyl ketones in *E. coli* where the paper title is Bacterial Ethyl Ketone Synthesis or Methyl Ketone Production in *E. coli* (Appl Environ Microbiol. 78, 70-80, 2012; Metab Eng. 26,

67-76, 2014; *Biotechnol Bioeng.* 115, 1161-1172, 2018). The detailed protocol was also written in the Methods (see Line 528-579).

Reviewer #1 major comment 4:

4. Results, Streptomyces albus engineering, paragraph 2: Since the ϕ C31r attP/int, VWB attP/int, and BT1 attP/int systems have different integration sites, introduction of three copies of the engineered PKS genes into the three different genomic loci is expected to enhance the production of the ketones. Please include this engineering strategy in the manuscript and describe the enhanced result.

Our response to reviewer #1 major comment 4:

We are confused by what the reviewer is asking. If the reviewer thinks that the best-producing *S. albus* strains contained three PKS genes, one in each integration site, the reviewer is incorrect. If, on the other hand, the reviewer believes that we put three copies of the PKS into each locus in separate strains and compared them, then the review is also incorrect. In fact, each *S. albus* strain contains only one PKS gene but at different integration loci (ϕ C31 attB and VWB attB sites). If the reviewer is suggesting construction of a *S. albus* strain that contains three PKS genes, one in each integration site, it is possible that the suggested strategy further improves the titers, which is important for industrial applications. However, introduction of multiple copies of the same PKS genes may cause genome recombination resulting in deletions or inversions, which is not favorable. To our knowledge, we have achieved the highest reported titer of any compound from any engineered PKS, so further engineering is actually beyond of our scope at this point.

Reviewer #1 m major comment 5:

5. Results, Short-chain ketone production from plant biomass: Use of condensers or trapping of the evaporated ketones using organic solvents is expected to overcome the troubles during the fermentation in the bioreactor. Please apply such process engineering strategies to prevent the loss of the ketones by evaporation and accurately examine the capacity of the engineered strains in the bioreactors.

Our response to reviewer #1 major comment 5:

As suggested by the reviewer, we examined whether organic solvents prevent ketone evaporation. We selected decane and dodecane as overlays. As shown in Fig. S9b,c organic solvents slightly prevented ketone evaporation from medium 042 (no cells) but more than 60% of ketone was evaporated after 24 h even under the best conditions tried (17% decane). We also tested if ALB188 can produce more ketones in medium 042 in the presence of decane or dodecane. Unfortunately, less ketone production was observed (Fig. S9d). According to the cell density measurements (Fig. S9e), decane and dodecane may be toxic to *S. albus*. We added this information in the main text (see Line 324-334, yellow highlighted). The reviewer is correct that condensers could be used to collect the ketones as they are produced. However, this is easier said than done. The water vapor must first be condensed by an initial condenser, and then a cold-trap (very cold) condenser can be used to collect the ketones. These set-ups are very difficult to build and control and would be well beyond the scope of this paper. The important

point is that even without condensers or traps our engineered PKS produced more product than any engineered PKS ever reported in the literature.

Reviewer #1 major comment 6:

6. Results, Short-chain ketone production from plant biomass: In this section, the authors supplemented amino acids for the production of different ketones. However, it should be discussed whether feeding expensive amino acids are economically feasible. Otherwise, microbial strains capable of producing these amino acids should be utilized as the base strains.

Our response to reviewer #1 major comment 6:

It is true that feeding amino acids would not be economically feasible to produce these specific ketones. We fed the amino acids so we could test how effective the engineered PKSs were to produce different ketones. As suggested by the reviewer, we included the point in the main text (see Line 299-301, yellow highlighted).

Reviewer #1 major comment 7:

7. Results, Short-chain ketones as gasoline oxygenates: Were the ketones used for the examination extracted from the cell culture? Please clearly mention how the ketones from the cell culture were extracted, and the percentages of the incorporated ketones from the cell culture.

Our response to reviewer #1 major comment 7:

We did not extract ketones from cultures because we needed to provide, at least, 500 ml of pure ketones. We purchased the corresponding ketones from vendors. We added the information to the Methods (see Line 687-689, yellow highlighted).

Reviewer #1 minor comment 1:

1. Introduction, paragraph 1: Please include proper citation of the report

Our response to reviewer #1 minor comment 1:

We added the citation (see Line 5-6, yellow highlighted).

Reviewer #1 minor comment 2:

2. Results, Rational PKS engineering, paragraph 1: Fig. 1 Fig. 1a

Our response to reviewer #1 minor comment 2:

As we revised Fig. 1 (as suggested by the reviewers 1 and 2), we no longer have Fig. 1a.

Reviewer #1 minor comment 3:

3. Results, Rational PKS engineering, paragraph 2: Please cite 'the recently reported PKS engineering strategy' in the first sentence

Our response to reviewer #1 minor comment 3:

As we revised Rational PKS engineering, we no longer have the paragraph. The paragraph was simplified and inserted into the Introduction as suggested by the reviewer 3 (see Line 60-72, yellow highlighted). The previous sentence includes the corresponding citations (see Line 57-60).

Reviewer #1 minor comment 4:

4. Results, Rational PKS engineering, paragraph 2: Supplementary Fig. 6 is mentioned before Supplementary Fig. 1 – 5. The same applies to the supplementary tables. Please reorder the supplementary materials so that the first item is mentioned first in the manuscript.

Our response to reviewer #1 minor comment 4:

As suggested by the reviewer, we reordered supplementary figures/tables.

Reviewer #1 minor comment 5:

5. Results, Rational PKS engineering, paragraph 3, 2nd sentence: Please cite appropriate reference supporting this sentence.

Our response to reviewer #1 minor comment 5:

The corresponding sentence was moved to the Discussion with a citation (see Line 439-443).

Reviewer #1 minor comment 6:

6. Results, Rational PKS engineering, paragraph 1, 2nd last sentence: Please cite appropriate reference supporting this sentence.

Our response to reviewer #1 minor comment 6:

As we revised Rational PKS engineering, we no longer have the paragraph. The paragraph was simplified and inserted into the Introduction as suggested by the reviewer 3 (see Line 60-72, yellow highlighted). The previous sentence includes the corresponding citations (see Line 57-60).

Reviewer #1 minor comment 7:

7. Results, Rational PKS engineering, paragraph 1, 2nd last sentence: Please cite appropriate reference describing the actinophage ϕ C31r attP sites and the corresponding integrase.

Our response to reviewer #1 minor comment 7:

We described the ϕ C31 attP/int system in the *Streptomyces* host selection, paragraph 2. We added the corresponding citation (see Line 141, yellow highlighted).

Reviewer #1 minor comment 8:

8. Results, Rational PKS engineering, paragraph 3: 8.8-fold instead of 9-fold.

Our response to reviewer #1 minor comment 8:

We revised it as suggested by the reviewer (see Line 201, yellow highlighted).

Reviewer #1 minor comment 9:

9. Fig. 1a: Please redesign the 'rational PKS engineering' part to display the strategy more clearly. It is suggested to explicitly describe the before and after constructs of the engineered PKS.

Our response to reviewer #1 minor comment 9:

We redesigned Fig. 1 as suggested by the reviewer.

Reviewer #1 minor comment 10:

10. Methods, Ketone production analysis: It is mentioned that since the authentic standard for 4-methyl-3-hexanone is not commercially available, 5-methyl-3-hexanone was used to quantify 4-methyl-3-hexanone. However, since the data obtained through this procedure might not be correct, the authors should also mention in the manuscript about this calculation method, such as '4-methyl-3-hexanone was quantified as 5-methyl-3-hexanone equivalent.

Our response to reviewer #1 minor comment 10:

We added the suggested sentence in the main text (see Fig. 2 legend, yellow highlighted).

Reviewer #2 (Remarks to the Author):

I was very impressed with the comprehensiveness and excellent quality of the work. This study nicely demonstrates how PKS engineering strategies, in combination with host, promoter and media optimization can significantly improve titers. The work establishes a strong platform for future metabolic and enzyme engineering involving modular PKSs towards the industrial manufacture of a diverse class of hydrocarbons. Nevertheless, I had some specific questions.

Reviewer #2 comment 1:

1. *I am a bit puzzled by the motivation that has been implicitly described in the manuscript for pursuing the synthesis of ketones. Since this has not been explicitly stated, the authors are running the risk of readers filling in the blanks themselves and this could detract from the impact of the work. My own interpretation of the reason that the authors are advocating microbial synthesis of short-chain ketones is that these molecules are presently manufactured from petroleum and that a good bio-based alternative could prove to be sufficient to meet their current demand. The authors also cite an ICIS report that discusses non-fuel uses of short-chain ketones. However, they then change their focus to using these molecules as fuel additives. Here too, the reasoning is a bit suspect (or incomplete, to be precise). While the octane number is an important criterion that is used to assess the performance of a fuel, it is not the only criterion. What about isothermal compressibility and auto-ignition temperature, which, according to fuel researchers, are equally or arguably more important than octane numbers. Why weren't these properties estimated? Additionally, since ethanol outperforms the methyl ketones in regard to octane numbers, it would be appropriate to discuss the 'blending' criteria that disqualify ethanol, and how the methyl ketones in question measure up. Physical reasons beyond octane numbers*

underpin EPA's 15% blending criteria for ethanol. What if blending of methyl ketones is similarly limited by physical incompatibilities?

Our response to reviewer #2 comment 1:

Short-chain ketones could be used in a wide variety of industrial applications (e.g. fuel, fuel additive, solvent, flavor, and fragrance) as we discussed in the Introduction. However, we agree that our current Introduction may detract from the impact of the work. We removed non-fuel uses for short-chain ketones from the Introduction and briefly stated their non-fuel uses in the Discussion (see Line 420-422, yellow highlighted). As suggested by the reviewer, we measured auto ignition temperatures of two major ketones produced from our engineered strains. We also measured octane numbers of pure 2-methyl-3-pentanone, which provided the exact same RON and MON as those of isooctane. We summarized the fuel properties of isooctane, common biofuels and our ketones in supplementary Table 7. Based on the data, we found that our ketones outperform common biofuels in terms of energy density, auto ignition temperature, and water solubility. We included the information in the main text (see Line 372-379, yellow highlighted).

Reviewer #2 comment 2:

2. Figure 1 is very similar to a figure from a previous study reported by the same authors. In fact, the only difference between the current figure and its previous version is the inclusion of a small PKS engineering diagram that is not very informative. I am not sure that the editorial policy for Nature Communications is but I am assuming that permission would be required to re-use part of a copyrighted work.

Our response to reviewer #2 comment 2:

We redesigned Fig. 1 as suggested by the reviewer.

Reviewer #2 comment 3:

3. How generalizable is the PKS engineering described by the authors to modular PKS that have been sourced from other strains such as Mycobacterium spp.? The lack of generalizability of PKS engineering across PKS, as well as hosts has hitherto been a significant hurdle for the field.

Our response to reviewer #2 comment 3:

The AT domain swapping method is generalizable. We have shown that it is possible to exchange ATs in several model PKS systems (ACS Synth Biol. 6, 139-147, 2017; ACS Chem Biol. doi: 10.1021/acscchembio.8b00422). We included the point in the Discussion (see Line 429-438, yellow highlighted). For the translational start site, it depends on the accuracy of the annotation of given PKSs. As we reported (ACS Chem Biol. 12, 2725-2729), different start codon prediction software predicted different start sites. Because there is no perfect software for high GC genomes at this point, experimentation may be necessary. We included the point in the Discussion (see Line 439-448, yellow highlighted).

Reviewer #2 comment 4:

4. I was very impressed by the results of the three PKS engineering methods detailed in the paper. The authors build a very compelling case for each method. However, these methods were tested under very different conditions and in a very different system from the system described in the current study. As a consequence, what the reader is being presented is actually the combined effect of many engineering strategies at once, along with improved host selection. If one were to perform a principal component analysis, what would the authors expect to see? Alternatively, if one has a hybrid PKS with known activity in *E. coli*, how much improvement can they hope to expect when the PKS is expressed in *S. albus*? An *in vitro* assay, or at least an assay in an *E. coli* host comparing improvements for all new engineering method combinations would be ideal.

Our response to reviewer #2 comment 4:

The reviewer is correct that we did not examine production between each change, nor did we do a principal component analysis to determine the impact of each change on production. However, this is easier said than done because, to test the generality of each method, we should test them using several model PKS systems that produce different products. The important point is that we achieved higher levels of product formation from an engineered PKS than have ever been reported in the literature by precisely engineering one specific PKS, along with host selection and engineering, which would provide a roadmap for PKS-based chemical production in a microbe. For *E. coli* vs. *S. albus*, it would be very difficult to answer the question at this point because they produce different substrates at different intracellular concentrations at different timings. To answer the question, we would need to know the above information as well as intracellular pH and ionic strength etc. because it is known that PKSs behave differently in different buffer conditions (Biochemistry 1996, 35, 2054-2060) and such a response would be different between PKSs.

Reviewer #2 comment 5:

5. The difference between LipPKS1+TE and the AT-swapped version is shown in Fig. S2-c, which is helpful. How much of the productivity enhancement comes from a less bulky group (H vs. methyl) near the reactive carbon?

Our response to reviewer #2 comment 5:

As we described in the main text (see Line 192-202), the AT-swapped PKS produced 3-hydroxyhexanoic acid from *n*-butyryl-CoA and malonyl-CoA with an approximately 10-fold faster k_{cat} compared to 2-methyl-3-hydroxyhexanoic acid production from *n*-butyryl-CoA and methylmalonyl-CoA by LipPks1+TE *in vitro* (ACS Synth Biol. 2017 6, 139-147). We also observed expected levels of 3-hydroxy acids *in vivo* (Fig. S5c).

Reviewer #2 comment 6:

6. It would be appropriate to see primers for engineering work carried out in this study (particularly, for the site modification to the ketoreductase) in the supplementary information.

Our response to reviewer #2 comment 6:

We provided the information in Supplementary table 8 and Supplementary methods (see Plasmids and strains)

Reviewer #2 comment 7:

7. The description of the culturing conditions is quite vague and unhelpful in its current form. For instance, on page 4, the authors state, "Each of the 12 recombinant strains was initially grown in TSB for a few days and then inoculated..." Could the authors be more precise? This information is critical for assessing productivity of the strains. In fact, since the authors are targeting fuel applications for the ketones, productivity is arguably the only metric that should be used to assess performance of the strains.

Our response to reviewer #2 comment 7:

We don't think the description of the culturing conditions is ambiguous. We provided the details in the Methods (see Line 529-580). In the main text, however, we described that each of the 12 recombinant strains was initially grown in TSB for "a few days". We also revised the sentence to be more precise (see Line 157).

Reviewer #2 comment 8:

8. On page 4, the authors state, "M042 was developed to maximize production of various secondary metabolites by Streptomyces and gave the best titers among several different media tested (data not shown)." Could this data be provided in the supplementary information? Was this data produced in the current study or a previous study? It would be helpful to see the relative performance, especially since the only difference between the two versions of M042 reported in the study is the replacement of sugars with plant biomass hydrolysates.

Our response to reviewer #2 comment 8:

We provided the data in Fig. S3.

Reviewer #2 comment 9:

9. Do Streptomyces spp. hosts naturally have an attB site or does it need to be introduced? If doing so was part of the work for this study, what locations were chosen and what primers/DNA was used to achieve this goal?

Our response to reviewer #2 comment 9:

We did not introduce an attB site for any integrase used in this study. Many *Streptomyces* strains carry attB sites for numerous actinophages.

Reviewer #2 comment 10:

10. Could the authors provide some LC-MS chromatograms and spectra in the supplementary information? How precise are the engineered PKS? Is there any by-product formation?

Our response to reviewer #2 comment 10:

We added representative LC-MS chromatograms in Fig. S7. As shown in Fig. S7, we observed expected ketone products only (no peak shoulders and split peaks). We also included this information in the main text (see Line 420-426, yellow highlighted).

Reviewer #2 comment 11:

*11. Could the authors provide more information about genetic engineering of *S. albus* in the form of protocols (if they were developed during the course of the current study) or citations?*

Our response to reviewer #2 comment 11:

We provided more information in the Methods (see Line 472-512, yellow highlighted).

Reviewer #2 comment 12:

12. In figure S5, which shows ketone production levels in a bioreactor, the maxima is achieved within the first 50 hours. However, in the biomass tests, sugar consumption occurs over 9 days. What is the fractional yield of the ketone from sugars and biomass, respectively? This is of particular interest for these strains' and PKSs' intended roles as providers of renewable energy and chemicals.

Our response to reviewer #2 comment 11:

As the reviewer pointed out, the production level in the bioreactor peaked sooner than that in the shake flask experiment. The reason is still unclear but one possible reason is that higher mass transfer of oxygen in the bioreactor results in more rapid conversion of sugars to 3-keto acids, which could be converted to the corresponding ketones at a faster rate because of more intense aeration and agitation. These conditions would also accelerate stripping of ketones from the fermentation broth, resulting in more rapidly decreasing titers once the ketone removal rate exceeds the production rate. We added these sentences in the main text (see Line 315-324, yellow highlighted). Regarding the yields, we initially calculated grams of product per gram of total sugar consumed (glucose + xylose, Fig. S6) and then calculated the yields (0.0125 g·g⁻¹ for ethyl ketones, 0.0028 g·g⁻¹ for methyl ketones). We then assumed that *S. albus* employs essentially the same pathways to produce the corresponding PKS substrates as model bacteria although its metabolism is not well studied. We assumed that 2 glucoses (1 for starter substrate, 1 for extender substrate) are required to produce 1 ketone product. Based on these assumptions, we calculated the approximate maximum theoretical yields. We provided the information in the main text (see Line 292-293 and Line 296-297, yellow highlighted).

Reviewer #3 (Remarks to the Author):

For this review, I prefer to send an anonymized report.

*This work demonstrated a route to produce various short-chain (C5-C7) ketones using hybrid PKSs in *Streptomyces*, and the titers and the following engine tests seem to have great potential for application.*

However, the concept has been proofed by previous report (ACS Synth Biol, 2017 6, 139-147). The Fig 1b and c in the current work is nearly the same as Fig. 4A in the previous article. If the novelty of this work is the improved yield, the authors should clearly clarify the novelty of the strategy employed and the detailed steps. Therefore, in my opinion, the design of this work seems lack of enough novelty required by Nat. Commun. But the results are impressive, so I suggest it be submitted to more specific journal.

Our response to reviewer #3 major comment:

We disagree with the reviewer that this work is not novel enough for publication in *Nature Communications*. We believe that it deserves publication in *Nature Communications* for the following reasons:

- We achieved higher levels of product formation from an engineered PKS than have ever been reported in the literature.
- We showed how several PKS engineering strategies could be used in combination to develop an engineered PKS that functions well in vivo.
- We produced unnatural ketones that have never been reported to be produced biologically.
- We tested the ketones as oxygenates in gasoline and demonstrated that they are as good as, if not better than, ethanol as an oxygenate.
- We achieved higher production levels of biogasoline candidates than have ever been reported in the literature using an engineered enzyme.

As pointed out by the reviewer, we previously reported short-chain ketone production *in vitro* and in *E. coli* (ACS Synth Biol. 6, 139-147, 2017). However, as we described in the main text, the production levels in *E. coli* were extremely low ($<5 \text{ mg}\cdot\text{l}^{-1}$) and we could not produce any ketone described in this paper because *E. coli* cannot produce the corresponding substrates for the PKSs employed via its native metabolism. It should also be noted that it is very common to see very low-level productions from engineered PKSs. For example, Menzella et al (Nat Biotechnol. 23 1171-1176, 2005) expressed 154 engineered PKS genes in *E. coli*. Although nearly half of them produced expected products, most of them gave very low titers ($<1 \text{ mg}\cdot\text{l}^{-1}$). The best titer was $23.5 \text{ mg}\cdot\text{l}^{-1}$. To overcome this issue, we believe that precise PKS engineering, well-considered host selection, host engineering, and medium optimization are necessary. In this paper, we further engineered the PKSs that are reported in the ACS Synth Biol paper. Based on our recent finding on translational start site of LipPks1 (ACS Chem Biol. 12, 2725-2729, 2017), we truncated the *N*-terminal tail. We also found that the catalytic Y to F mutation in the KR domain increases ketone production capability (Fig. S1b). In addition, we compared 4 different *Streptomyces* strains and engineered the most promising host, *S. albus* (Fig. 2a-f). We tested several different media and found that medium 042 is the excellent polyketide production medium in *S. albus* (Fig. S3), which has not been reported. Further, we achieved expected ketone production from plant biomass with significantly increased titers (a titer exceeding $1 \text{ g}\cdot\text{l}^{-1}$), which has never been achieved by any group using any engineered PKSs. We also first demonstrated that short-chain ketones could be blended to gasoline to increase the octane numbers (Fig. 3). We also measured other fuel properties of ketones produced from our strain (Table S7), which indicates that short-chain ketones could also work as a gasoline replacement.

In addition, as decried in the Discussion, we achieved higher levels of biogasoline production than any microbial strain that encodes engineered enzymes (see Line 389-398): 728 mg·l⁻¹ of short-chain alcohol (Nat Commun. 6, 10005, 2015); 581 mg·l⁻¹ of short-chain alkane (Nature. 502, 571-574, 2013). We strongly believe that our results are very novel and deserve publication in *Nature Communications*.

Reviewer #3 other comment 1:

(1) Too many background information in the first part of results "Rational PKS engineering", Moving this words to Introduction should be better.

Our response to reviewer #3 other comment 1:

As suggested by the reviewer, we moved the background information in Rational PKS engineering to Introduction.

Reviewer #3 other comment 2:

(2) In the test, Supplementary Fig. 6 appears first. The authors should order the figures in sequence.

Our response to reviewer #3 other comment 2:

We received the same comment from reviewer 1 (minor comment 4). We fixed the issues as described above.

Reviewer #3 other comment 3:

(3) The authors only test three promoters in this work, should other promoters be better? To coordinate the biosynthesis of short-chain ketones and other physiological events within the hosts, fine-tuning the expression using a panel of promoters with different profiles of time and strength, I think, will be helpful.

Our response to reviewer #3 other comment 3:

It is possible that the suggested strategy further may improve the titers. However, we have already achieved the highest titer of any compound from any engineered PKS, so this is actually beyond our scope at this point. Further engineering of the host to yield incremental titer increases would be something that companies interested in commercial production would do.

Reviewer #3 other comment 4:

(4) For rational engineering, two chromosomal locations were tested, it seems a little subjectivity. Might other chromosomal loci be better?

Our response to reviewer #3 other comment 4:

As described in the main text, we also tested BT1 attP/int. However, the frequency of obtaining transconjugants of *S. albus* with a control vector containing BT1 attP/int was quite low. For this reason, we employed ϕ C31 attP/int and VWB attP/int to generate *S. albus* strains that encode desired PKSs. We agree that the other chromosomal sites might further improve the titers.

However, as we mentioned above, we achieved the highest titer of any compound from any engineered PKS and have demonstrated the feasibility of employing this system for industrial use.

Reviewer #3 other comment 5:

(5) When using plant biomass, the utilization levels and productivities of the engineered strains were the more concerned data. However, the author did not provide the information.

Our response to reviewer #3 other comment 5:

We had essentially the same comment from Reviewer 2 (comment 11). Please check our response to the comment.

Reviewers' comments:

Reviewer #1 (Remarks to the Author):

Authors revised the manuscript appropriately, and I do not have any further comment.

Reviewer #2 (Remarks to the Author):

I am very satisfied with the authors' responses to my questions. They have suitably modified the manuscript and the revisions have greatly improved the scope and impact of the work. I do not have any additional comments or questions.

Reviewer #3 (Remarks to the Author):

This manuscript entitled "Microbial production of short-chain ketones using hybrid polyketide synthases" puts forward a significant approach to produce ketones that play key roles in fuels. It is well-known that polyketide synthases are multi-domain enzymes that resemble a modular assembly line for secondary metabolites biosynthesis, but it is truly a promising idea to engineer them for various short-chain (C5-C7) ketones. Although the concept has been proved by previous report (ACS Synth Biol, 2017 6, 139-147), the authors claim the unprecedented production in *Streptomyces* species and this higher production levels is of great significance for microbial production of biogasolines. The authors seem to address all the comments the reviewers concerned. So, I believe this work presents us a new route for the production of biofuels and is worth being published after revision.

Comments:

1. PAL1 (Fig. 1) is very crucial when you design these enzymes in this work. However, I have trouble finding the details for PAL in your manuscript, as well as the reference. Please give the readers more information.
2. Since the detection methods of products could not be accurate due to evaporation of short-chain ketones, we suggest you use GC-MS relative approaches to quantify the gas emission of bioreactor. By this way, we can make the solid conclusion whether the evaporation of target ketones prevented the use of the bioreactor. Only the decrease of short-chain ketones in the medium (Supplementary Fig. 8) is not sufficient to conclude that ketone removal rate exceeds the production rate after a certain period of time.
3. Whatever, the concept is much similar to the previous paper (ACS Synth Biol, 2017 6, 139-147), which firstly demonstrated some short-chain ketones production as this work displays. In addition, the fact that truncated N-terminal tail of LipPks1 can improve titer has been published (ACS Chem Biol. 2017 Nov 17;12(11):2725-2729). Therefore, I suggest this manuscript should lay more emphasis on your novel strategy for yield improvement in *Streptomyces* and the potential application of ketones produced.

Reviewers' comments:

Reviewer #1 (Remarks to the Author):

Authors revised the manuscript appropriately, and I do not have any further comment.

Thank you.

Reviewer #2 (Remarks to the Author):

I am very satisfied with the authors' responses to my questions. They have suitably modified the manuscript and the revisions have greatly improved the scope and impact of the work. I do not have any additional comments or questions.

Thank you.

Reviewer #3 (Remarks to the Author):

This manuscript entitled “Microbial production of short-chain ketones using hybrid polyketide synthases” puts forward a significant approach to produce ketones that play key roles in fuels. It is well-known that polyketide synthases are multi-domain enzymes that resemble a modular assembly line for secondary metabolites biosynthesis, but it is truly a promising idea to engineer them for various short-chain (C5-C7) ketones. Although the concept has been proved by previous report (ACS Synth Biol, 2017 6, 139-147), the authors claim the unprecedented production in Streptomyces species and this higher production levels is of great significance for microbial production of biogasolines. The authors seem to address all the comments the reviewers concerned. So, I believe this work presents us a new route for the production of biofuels and is worth being published after revision.

Thank you. We agree.

Comments:

1. PAL1 (Fig. 1) is very crucial when you design these enzymes in this work. However, I have trouble finding the details for PAL in your manuscript, as well as the reference. Please give the readers more information.

Good catch. Thank you for this comment. We have added the information to the revised manuscript Fig. 1 (yellow highlighted).

2. Since the detection methods of products could not be accurate due to evaporation of short-chain ketones, we suggest you use GC-MS relative approaches to quantify the gas emission of bioreactor. By this way, we can make the solid conclusion whether the evaporation of target ketones prevented the use of the bioreactor. Only the decrease of short-chain ketones in the medium (Supplementary Fig. 8) is not sufficient to conclude that ketone removal rate exceeds the production rate after a certain period of time.

The reviewer is incorrect in several ways.

First, the reviewer is incorrect in stating that evaporation of target ketones prevented the use of the bioreactor. We showed results from both the bioreactor cultures and from the shake flask cultures. The concentrations of ketones in the liquid phase of the bioreactors were lower than those in the liquid phase of the shake flasks. This is not surprising given the differences between bioreactors and shake flasks. Bioreactors differ from shake flasks in that gas (in our case air) is blown through the bioreactor, which strips any volatile compounds from the liquid phase faster than does the stagnant gas phase above the liquid in a shake-flask. As a result, the concentration of any volatile compound (in our case the ketones) in the liquid phase of a bioreactor will be less than the concentration of the volatile compound in the liquid phase of a shake flask. If one assumes that the specific rate of production of the ketone (amount of ketone produced per unit time per unit biomass) is the same in shake flask cultures and bioreactor cultures, then for the same amount of biomass the concentration of ketones in the aqueous or gas phase of the bioreactor will be less than the concentration of ketones in the aqueous or gas phase of the shake flask. However, we cannot assume that the specific rate of production of the ketones from biomass grown in bioreactors is the same as that from the biomass grown in shake flasks. Most likely the specific production rates are different.

Second, the reviewer is incorrect in stating that GC-MS measurement of the concentration of ketones in the reactor outgas will give anything different from our LC-MS measurement of the concentration of ketones in the liquid phase. The concentrations of ketones in the liquid phase can be assumed to be in equilibrium with the concentrations of ketones in the gas phase. So, the measurement by LC-MS (which we measured) and GC-MS (which the reviewer suggested) are no different. They are both concentration measurements, just of different phases that are in equilibrium. One could continuously measure the concentration of the ketone in the gas phase coming out of a bioreactor (using an inline MS detector that would continuously measure the concentration of the ketones in the gas coming out of the bioreactor), and knowing the gas flow rate calculate the total amount of ketone stripped out of the reactor over the course of the experiment. We attempted to do this but could not with the inline MS detector available to us.

Third, the reviewer is incorrect that we cannot make the case that the rate of removal exceeds the rate of production in a certain phase. In Supplementary Figure 8, we show that the concentration in the liquid phase increases until approximately 36 hours, at which time the concentration decreases. The gas flow rate in the bioreactor was constant during the entire run. Assuming that the volatilization of the ketones is the primary means of removal of the ketones from the bioreactor (e.g., being stripped out by the air blown through the bioreactor) and that the removal rate does not change significantly throughout the entire run, then one can conclude that the production exceeds loss prior to 36 hours and that loss exceeds production after 36 hours.

3. Whatever, the concept is much similar to the previous paper (ACS Synth Biol, 2017 6, 139-147), which firstly demonstrated some short-chain ketones production as this work displays. In addition, the fact that truncated N-terminal tail of LipPks1 can improve titer has been published (ACS Chem Biol. 2017 Nov 17;12(11):2725-2729). Therefore, I suggest this manuscript should

lay more emphasis on your novel strategy for yield improvement in *Streptomyces* and the potential application of ketones produced.

The reviewer made the same comment in the previous version of the comments. In response to these comments, we significantly revised the manuscript to emphasize *in vivo* production and only included PKS engineering strategies that impact *in vivo* production of ketones. As such, we believe that these comments are inaccurate for the revised manuscript.

First, while the cited papers did demonstrate very low ketone production and the importance of truncating the N-terminus of the LipPks1, putting all of these elements together allowed us to achieve the high-level production of ketones. In fact, these levels are higher than have ever been reported for *in vivo* or *in vitro* production of a polyketide using an engineered polyketide synthase. We revised Figure 1 in a way so that our methodology could be used for improving production of polyketides using engineered hybrid polyketide synthases inside cells.

Second, the revised version of the manuscript emphasized *in vivo* production of ketones using *Streptomyces*, which is evident in the figure count and in the text. Of the three figures in the revised manuscript, Figure 1 shows the methodology for improve the engineered PKS so that it would express better and have better activity in *Streptomyces* and Figure 2 shows the *in vivo* ketone production for all of the various changes to the PKSs. The only other figure in the main text, Figure 3, shows the data for ketone impact on RON and MON. Fully 2 out of the 3 figures are for *in vivo* production. Of the supplementary figures, 7 of the 9 figures have data for *in vivo* production of ketones or for consumptions of substrates by the engineered *Streptomyces*. Figure S2 shows construction of the engineered *Streptomyces* strains. Figure S4 shows *in vitro* ketone production. So, as measured by the Figures, only three of the twelve figures do not have any data or relationship to *in vivo* production of ketones using an engineered hybrid PKS.

In terms of text, only first part of paragraph 4 (lines 56 to 74) and paragraph 5 (lines 85 to 107), 40 lines total, emphasized construction of the hybrid PKS. All other text directly addressed *in vivo* production of ketones using our engineered hybrid PKS or decarboxylation of ketoacids to ketones or to the use of ketones as additives to gasoline. By count of lines, only 40 lines or 7% of the text focuses on engineering the PKS; the other 93% focuses on *in vivo* production of ketones using *Streptomyces*. It would be very difficult to remove these 40 lines and have an understandable manuscript that others could use to engineer PKSs for use *in vivo*.